# Identification of Incomplete Annotations of Biosynthesis Pathways in Rhodophytes Using a Multi-Omics Approach

**DOI:** 10.3390/md22010003

**Published:** 2023-12-19

**Authors:** Lachlan J. McKinnie, Scott F. Cummins, Min Zhao

**Affiliations:** 1Seaweed Research Group, University of the Sunshine Coast, Maroochydore, QSL 4558, Australia; lachlan.mckinnie@research.usc.edu.au (L.J.M.); scummins@usc.edu.au (S.F.C.); 2School of Science, Technology, and Engineering, University of the Sunshine Coast, Maroochydore, QSL 4558, Australia; 3Centre for Bioinnovation, University of the Sunshine Coast, Maroochydore, QSL 4558, Australia

**Keywords:** Rhodophyta, metabolism, phytosterol, gene duplication, bioinformatics, haem peroxidase

## Abstract

Rhodophytes (red algae) are an important source of natural products and are, therefore, a current research focus in terms of metabolite production. The recent increase in publicly available Rhodophyte whole genome and transcriptome assemblies provides the resources needed for in silico metabolic pathway analysis. Thus, this study aimed to create a Rhodophyte multi-omics resource, utilising both genomes and transcriptome assemblies with functional annotations to explore Rhodophyte metabolism. The genomes and transcriptomes of 72 Rhodophytes were functionally annotated and integrated with metabolic reconstruction and phylogenetic inference, orthology prediction, and gene duplication analysis to analyse their metabolic pathways. This resource was utilised via two main investigations: the identification of bioactive sterol biosynthesis pathways and the evolutionary analysis of gene duplications for known enzymes. We report that sterol pathways, including campesterol, β-sitosterol, ergocalciferol and cholesterol biosynthesis pathways, all showed incomplete annotated pathways across all Rhodophytes despite prior in vivo studies showing otherwise. Gene duplication analysis revealed high rates of duplication of halide-associated haem peroxidases in Florideophyte algae, which are involved in the biosynthesis of drug-related halogenated secondary metabolites. In summary, this research revealed trends in Rhodophyte metabolic pathways that have been under-researched and require further functional analysis. Furthermore, the high duplication of haem peroxidases and other peroxidase enzymes offers insight into the potential drug development of Rhodophyte halogenated secondary metabolites.

## 1. Introduction

Rhodophytes (red algae) form a globally distributed and genetically diverse phylum of aquatic organisms that produce a wide variety of bioactive compounds. There are over 7000 known species of red algae [1], which include both red seaweeds (macroalgae) and microalgae. Phylogenetically, Rhodophyta is divided into seven classes. The red seaweeds belong to the classes Bangiophyceae and Florideophyceae, which include the edible cultured seaweeds *Porphyra yezoensis*, *Chondrus crispus*, and *Kappaphycus alvarezii*. The remaining five classes are composed of red microalgae and include Cyanidiophyceae, Compsopogonophyceae, Porphyridiophyceae, Rhodellophyceae, and Stylonematophyceae. These include the cultured polyextremophilic genera *Galdieria* and *Cyanidioschyzon* [2], as well as the commercially cultivated *Porphyridium* [3]. 

Rhodophyte macroalgae are commonly cultured for the production of multiple widely used compounds, such as carrageenan, agar, and alginate [4], while microalgae such as *Porphyridium* are cultured for metabolites such as phycoerythrin [3]. They are increasingly also a target for marine biodiscovery in a variety of fields, including medicine, nutrition, material science [5], and climate change mitigation [6,7]. Of the bioactive compounds produced by red algae, phytosterols show clear medicinal benefits. In contrast to other algae, Rhodophyte sterol content is predominantly cholesterol [8,9]; however, red algae have been shown to possess small amounts of C24 methyl- and ethylsterols [10]. These compounds have been shown to have a variety of pharmacological and nutraceutical benefits, including anti-cancer, anti-diabetes, and anti-inflammatory activity [11,12]. 

Red seaweeds, particularly the Florideophytes, also produce a wide range of halogenated natural products with potential medical and industrial applications, with brominated terpenes and halomethanes being particularly common [13,14]. The biosynthesis of halogenated metabolites is reliant on numerous enzymatic reactions and biosynthetic genes. Of these, heme peroxidases and related enzymes are well known to be important enzymes in the biosynthesis of halogenated metabolites [15] and have been observed to be duplicated in *C. cripsus* [16]. Gene duplication is a major contributor to evolution and is a known source of gene subfunctionalisation [17].

Despite their importance, Rhodophyte omics have been relatively underdeveloped, particularly in regard to published whole genome and transcriptome assemblies, with relatively few published and even fewer annotated [18,19]. However, recent years have seen a sharp increase in the number of Rhodophyte assemblies published. In particular, several studies have published large datasets that enable broad comparisons of Rhodophyte assemblies. A 2019 study by Rossoni et al. [2] published 10 genome assemblies from the class Cyanidiophyceae, with their online dataset including protein predictions and functional annotations [20]. Likewise, large transcriptome projects have greatly increased the number of Rhodophyte assemblies available. The Marine Microbial Eukaryotic Transcriptome Sequencing Project (MMETSP) [21] published over 600 transcriptomes, including ten from Rhodophyte microalgae, while the One Thousand Plant Transcriptome Project (1KP) [22] released 28 Rhodophyte transcriptome assemblies with protein annotations in their dataset [23]. Furthermore, several important Rhodophytes that have historically not had much genomic investigation have had assemblies published recently. For example, the red seaweed *Asparagopsis taxiformis*, which has been studied as a source of bioactive, only had a full genome assembly published in 2020 [14], while several assemblies from edible seaweeds, including *Gracilaria* and *Neoporphyra* sp. were only published within the last five years.

These large datasets of genome and transcriptome assemblies can be used to reconstruct and analyse the metabolic pathways of a wide range of organisms that otherwise have little omics data available. Key resources such as the KEGG database enable metabolic reconstruction using genomic or transcriptomic data. Whole-genome and transcriptome data also allow for the integration of metabolic reconstruction with phylogenomics in order to understand how an organism’s metabolic structure is reflected in its evolutionary history. This can be inferred using gene orthologue and gene duplication prediction via tools such as OrthoFinder [24]. This can be combined with other phylogenomic investigations, such as inferring genome expansion using gene duplication analysis and the prediction of repetitive elements.

Thus, this study aimed to utilise the new range of Rhodophyte genome and transcriptome assemblies to develop a resource for the in silico metabolic reconstruction of red algal metabolic pathways, using organisms representing all seven Rhodophyte classes. Furthermore, we aimed to combine phylogenetic inference and gene duplication analysis with functional annotation to infer how different functional characteristics are distributed across their evolutionary history. This study used a broad approach to bioinformatics in order to perform a large-scale analysis of red algal metabolic trends. This included organisms that have until recently been relatively under-researched, as well as the incorporation of newly published datasets across both genomes and transcriptomes.

## 2. Results

### 2.1. Data Collection and Functional Annotation

Data collection for red algal assemblies returned 34 genome assemblies from the NCBI database, as well as an additional *A. taxiformis* SC genome that has since been published on NCBI (GCA_030407315.1) [25]. Searching for Chlorophyta returned 170 assemblies, of which 41 had protein annotations available. As these were all green microalgae, two unannotated green macroalga assemblies (*Ulva mutabilis* and *U. prolifera*) were downloaded for annotation. A subset of 4 Chlorophytes (both *Ulva* genomes, *Ostreococcus tauri*, and *Chloropicon primus*) were chosen for functional annotation, while the remaining assemblies only had general statistics recorded. Searching for Glaucophyta returned only a single assembly (*Cyanophora paradoxa*), which did not have protein annotations available.

Genome assemblies were complemented with transcriptome assemblies sourced from the MMETSP and 1KP project databases, from which 10 and 28 Rhodophyte assemblies were downloaded, respectively. The MMETSP transcriptomes had assemblies but not protein annotations available, while the 1KP transcriptomes had annotated protein and CDS sequences available. In total, 72 Rhodophyte assemblies (34 genomes, 38 transcriptomes) were downloaded (Figure 1A), representing 46 species across all seven classes (Appendix A). One genome, *G. sulphuraria* 107.79, was immediately discarded due to extremely low assembly quality.

Analysis of repeat masking revealed highly variable results across red algae, with results ranging from 2.65 to 71.7% (mean = 28.96, SD = 19.2%), and a Levene’s test showed significant variation when grouped by class (*p* = 1.6 × 10^−3^). However, despite the high variability of masking values, a significant difference was observed between classes (*p* = 3.28 × 10^−5^, F = 11.74; Figure 1B). Median repetitive regions between classes were 41.57% for Bangiophyceae and 54.14% for Florideophytes, while Cyanidiophyceae and Porphyridiophyceae had medians of 20.01% and 4.20%, respectively. The composition of repeats also varied between classes. Most Florideophytes showed higher proportions of retroelements, DNA transposons, and unclassified repeats. Bangiophytes were similar to Florideophytes but typically had more simple repeats. Cyanidiophytes were split between the Cyanidioschyzonales and the Galdieriales plus *C. caldarium*. The Cyanidioschyzonales showed a fairly even mix of unclassified repeats and DNA transposons, while the Galdieriales showed mostly unclassified repeats, with some *Galdieria* assemblies also showing some retro elements. Two of the *Porphyridium* assemblies showed repeat distributions containing mostly DNA transposons, simple repeats, and unclassified repeats, but *P. purpureum* CCMP1328 also possessed a higher proportion of retroelements, as well as a higher total percentage of repeats (13.99%, compared to 3.3–4.2%). This corroborates previous evidence of genome expansion in Rhodophytes, particularly in the red seaweeds [26], and is likely linked to increased gene duplication.

A total of 15 genomes and all 10 MMETSP transcriptomes were annotated with AUGUSTUS. Protein BUSCO scores (Figure 1C) ranged from 20.4 to 94.1% (mean = 62.2%). There was no significant difference in BUSCO scores between assemblies with prior protein annotations and those annotated as part of this study (*p* = 0.81, t = −0.24, df = 45), indicating high enough quality to provide reliable, functional annotation. By contrast, the transcriptome assemblies used had lower BUSCO scores than the genomes (means = 41.29 ± 1.91 and 40.18 ± 5.57 for the 1KP and MMETSP transcriptomes, respectively), while the Chlorophyte assemblies had a higher average BUSCO score (mean = 77.83 ± 2.08). A total of 31 Rhodophyte genome and 27 transcriptome assemblies met the BUSCO completeness threshold of ≥35% complete and ≤50 missing. These assemblies were then used for comparative functional annotations. Assemblies that did not meet the criteria were also annotated but were excluded from further analysis.

Protein counts (Figure 1D) ranged from 3841 to 49,327. *R. marinus* MMETSP0011 was discarded as an outlier, as it had over 40,000 protein annotations. There was a noticeable difference in protein counts between the genome sequences and the 1KP transcriptomes, with the 1KP transcriptomes having significantly fewer proteins than the genomes (*p* = 3.22 × 10^−6^, t = 5.45), while the MMETSP transcriptomes had a more comparable number of protein sequences (*p* = 0.25, t = −1.2). Mean protein length (Figure A1) was largely consistent between Rhodophyte classes in the genome assemblies, though the *P. purpureum* assemblies showed a higher mean protein length (mean 518 amino acids (aa)). There was however a significant difference in mean protein length between the Rhodophyte genome and transcriptome assemblies (*p* = 1.95 × 10^−21^, df = 63, t = 14.29), with the 1KP and MMETSP transcriptomes having lower mean protein lengths (293.6 and 321.5 aa, respectively). The Chlorophyte genomes used for comparison had mean protein lengths ranging from 273.1–1035.4 (mean 526.8), which was significantly higher than the red algal genomes (*p* = 0.004, t = −3, df = 55). Overall, the genome-based proteomes were more comprehensive than the transcriptomes, with Chlorophytes more complete than Rhodophytes. The MMETSP transcriptomes were larger than the 1KP transcriptomes but, on average, had slightly lower BUSCO completeness scores. These annotation statistics were used to guide the analysis methods. As the transcriptome assemblies had notably different characteristics, further analyses were split between genome classes and transcriptome sources (1KP and MMETSP transcriptomes).

A total of 64 assemblies, comprising 36 genomes and 28 transcriptomes, passed the BUSCO quality filter. Inference of an approximate phylogeny with OrthoFinder v.2.5.4 [24,27,28] (Figure 2) inferred a total of 25,079 orthogroups, from which 13,612 gene trees were resolved and a consensus species tree inferred. Phylogeny largely followed established phylogenetic structures, but incongruities were observed in the Cyanidiophytes, with *C. caldarium* UTEX2393 clustering with *G. sulphuraria* with *G. phlegrea* clustered outside. *C. yangmingshanensis* also clustered with *C. merolae* Soos outside of the other *C. merolae* assemblies. This was supported using phylogeny derived from universal orthologues using BuscoPhylo [29] (Figure A2) and corresponded to plastid gene trees inferred by Park et al., 2023 [30].

### 2.2. Metabolic Reconstruction

Metabolic reconstruction with the KEGG and eggNOG annotations returned a total of 410 pathway modules (Figure 2, Appendix A), of which 384 were pathway modules and 26 were complex modules. A total of 44 pathway modules and 21 complex modules were marked as absent but had at least one KO present, signifying a partial block in the module; these were graded using the same standard but using the total number of KO “steps” instead of blocks and incorporated with the other results. In total, 37 modules were complete in all assemblies, though only 3 when using the strict standard, while 74 pathways were absent in all assemblies, largely representing bacterial and fungal metabolite biosynthesis pathways. 

The number of complete modules ranged from 79–138 (median = 111) (Figure 2). The Rhodophyte genomes and the MMETSP transcriptomes had, on average, more complete pathways (mean = 120.4 and 115.4, respectively), while the 1KP transcriptomes had fewer (102.9). No significant difference was observed in the Rhodophyte genomes (*p* = 0.525, F = 0.762, df = 30) nor when the MMETSP transcriptomes were grouped with them (*p* = 0.491, F = 0.873, df = 35), but showed significant differences when grouped with the 1KP transcriptomes (*p* = 1.8 × 10^−6^, F = 11.06, df = 53), indicating a loss of identifiable metabolic genes in the 1KP transcriptomes.

#### 2.2.1. Sterol Pathways Were Annotated as Incomplete across Algal Assemblies

To explore the functional annotation data further, a medically significant metabolic pathway was selected for further investigation. The phytosterol biosynthesis module M00917 represents the biosynthesis pathways for the phytosterols campesterol and sitosterol from squalene 2,3-epoxide. Both pathways are largely identical from a gene orthologue context, with the sitosterol pathway only having one additional orthologue compared to the campesterol pathway, though the reactions differ.

All individuals showed an incomplete phytosterol pathway module; however, all assemblies reported at least four of the KEGG orthologues required, with a median of eight orthologues present across all assemblies. Overall, consistent patterns were observed between assemblies of different red algal classes. In terms of similarities between classes, all assemblies bar one showed the initial orthologue, K01853 (cycloartenol synthase), while the majority of assemblies across all classes showed at least two of three orthologues in the middle of the pathway (K08246, K09717 and K00222). All red algal genome assemblies were missing orthologue K08242, the only orthologue unique to sitosterol, which encodes a 24-methylenesterol C-methyltransferase, though it was present in the outgroup and in three of the 1KP transcriptome assemblies. Differences were also observed between classes. An orthologue encoding sterol 24-C-methyltransferase (K00559) was absent in the Compsopogonophyceae, Bangiophyceae, and most of the Florideophyceae assemblies but present in the remaining Rhodophyte classes, while the orthologue K01824 (cholestenol Delta-isomerase) was present only in the Cyanidiophyceae and some outgroup assemblies. Cyanidiophyceae also showed further differences, with K00213 (8-dehydrocholesterol reductase) absent in Cyanidiophyceae but not in other classes, while K0928 (Delta 24-sterol reductase) was absent in the Galdieriales and *C. caldarium*, but present in the Cyanidioschyzonales and the other Rhodophyte classes.

Similar to the phytosterol module, the cholesterol and ergosterol biosynthesis modules, M00101 and M00102, respectively, were largely annotated as incomplete across the red algal assemblies. No assemblies had any completion reported for the cholesterol biosynthesis module; however, 24 out of 64 were reported as having two blocks missing, including four out of five Porphyridiophyceae, *R. violacea*, and the Bangiophyceae assemblies. Interestingly, the Cyanidioschyzonales were reported as two blocks missing, but not the Galdieriales. As for ergocalciferol, only three were reported as two blocks missing, including both the Stylonematophyceae assemblies; however, all three *P. purpureum* genomes were reported as having loosely complete phytosterol pathways. This contradicts in vitro evidence of sterol production in red algae, indicating that red algae may have novel genes involved in the biosynthesis of sterols that have not been identified in the KEGG database. Evidence of alternative sterol biosynthesis genes has been found in other organisms. For example, alternative squalene epoxidase genes, which are key to steroid production in eukaryotes, have been shown to have alternative genes not previously identified in diatoms and Cryptophyceae [31]. Furthermore, Belcour et al., 2020 [32] describe an alternate cholesterol biosynthesis pathway for *Chondrus crispus* predicted using in silico analysis that differs from the pathway used by the KEGG database.

#### 2.2.2. Highest Variation in Pathway Completion between Classes Was Observed in Amino Acid and Vitamin/Cofactor Biosynthesis Pathways and Showed Differences in Urea Pathways in Cyanidiophytes

Further comparison of metabolic pathways by their rates of completion (Figure 3) showed that overall pathways were conserved between Rhodophyte classes in the genome assemblies (*p* = 0.54, F = 0.72, df = 783) but that there was significant variation when the transcriptomes were also included and grouped by source (*p* = 0.036, F = 2.25). Most modules showed little variation between Rhodophyte classes, with 40 modules showing no variation and 138 showing a variance of less than 0.2. The most variable modules (Figure 3A; standard deviation ≥ 0.2, *n* = 58) showed significant differences between Rhodophyte classes (*p* = 0.023, F = 3.095, df = 231), as well as when the transcriptomes were included, grouped by source (3.79 × 10^−6^, F = 6.89, df = 347). Overall, however, most pathways were conserved, with an overlapping of the modules found in 50% of genomes for each class, showing 98 complete pathways (64%) across all four classes (Figure 3B). 

Of the most variable pathways, amino acid metabolism and vitamin/cofactor metabolism modules were the most prevalent, with each representing approximately a quarter of the modules, followed by carbohydrate and energy metabolism modules, each with seven modules present. Amino acid pathways predominantly belonged to the aromatic amino acid module category, while carbohydrate modules present were entirely from the module category “Other carbohydrate metabolism”, with none from the “central carbohydrate metabolism” module class. The remaining modules represented glycan and lipid metabolism (five each), nucleotide metabolism (three modules), and terpenoid/polyketide biosynthesis modules with two modules. This was generally reflective of the distribution of module classes in the KEGG database but showed variation from it in terms of module categories. 

These variable pathways also showed variation within classes, including notably within Cyanidiophyceae (Figure 3C). Patterns of module completion and orthologue presence or absence aligned with taxonomic order across several modules, indicating differences between the Galdieriales and the Cyanidioschyzonales (Appendix A), such as in the module M00555 (betaine biosynthesis), which was complete in the Galdieriales, but incomplete in the Cyanidioschyzonales, or in the two ascorbate biosynthesis pathways (M00114 and M00129), which showed complementary patterns of orthologue presence and absence in the orthologues K00225 and K00103. This showed distinct differences in the urea utilisation pathways between Galdieria and *Cyanidioschyzon*, with *Galdieria* showing far more complete urea-associated pathways (M00565 and M00959), with the *Cyanidioschyzon* assemblies not showing any complete blocks from M00546 at all. Correlation analysis showed specific gene orthologue or block presence correlated with taxonomic order across all eight pathways, though most genes and blocks were conserved (Table 1). Intraspecific variation was observed within the *Galdierias*, whereas the *Cyanidioschyzonales* showed more uniformity. This variation correlated with BUSCO scores in the *Galdieria* assemblies and in *Cyanidium,* implying greater completion rates of these pathways in the *Galdieria* and *Cyanidium* assemblies than the annotations inferred. No BUSCO quality correlation was observed in the Cyanidioschyzonales for these pathways. 

### 2.3. Phylogenomic Investigation Reveals Large Duplication of Haem Peroxidases in Florideophyte Algae

The relationship between metabolic activity and evolutionary history was further investigated using gene duplication analysis with a second run of OrthoFinder v.2.5.4 using the genome assemblies. A total of 22,797 orthogroups were constructed, and 12,512 gene trees were inferred. A species tree was constructed using 353 single-copy orthologues, of which three were universal. The species tree was verified by inferring a maximum-likelihood tree using the same sequence alignment via RAxML-NG [33], which returned the same tree with 100% bootstrapping on all nodes. Gene duplications were extracted from OrthoFinder v.2.5.4 (Appendix A), and the duplicated genes of each major phylogenetic node had their InterPro (Figure 4A, Appendix A) and GO annotations (Figure 4C, Appendix A) integrated with the nodes by correlating duplicated genes with their functional annotations (Appendix A). 

Total gene duplications for each assembly node (Figure 4B) were highest in the Florideophytes, Bangiophytes, and the outgroup and lowest in the *Porphyridium* and Cyanidiophyte assemblies. Within the Cyanidiophytes, duplications were significantly higher in the Galdieriales than in the Cyanidioschyzonales (*p* = 0.002, t = −4.37). *C. caldarium* showed lower rates of gene duplication, closer to the Cyanidioschyzonales. Duplicated gene annotations showed that many functions and enzyme annotations were conserved between taxa. Core InterPro enzymes were common across phylogenetic nodes, as were GO functions such as ATP and protein binding. However, specific functions and enzymes were observed to have differences between taxa. Cyanidiophyceae showed a higher rate of duplication of transmembrane transporters, most prominently the major facilitator superfamily (MFS) transporters, and was significantly higher in the Galdieriales and *C. caldarium* than in the Cyanidioschyzonales. In the Florideophytes, haem peroxidases were highly duplicated, particularly in the Gracilariales and in *Asparagopsis*. The Gracilariales and *Asparagopsis* also showed a notable duplication of the von Willebrand factor domain (IPR002035), as observed in the InterPro domains (Figure A3), while *Porphyridium* showed a large duplication of the integrase catalytic core domain (IPR001584).

## 3. Discussion

### 3.1. This Resource Covers a Wide Range of Red Algae, but Many Are Still Underrepresented

As research and development on red algae continue to grow, there will be an increasing need for strong omics resources that can help explain the biological questions surrounding Rhodophyte metabolism, culturing, and other facets of their biology. Recent sequencing has led to a drastic increase in the number of Rhodophyte genome assemblies available, and it is likely that this will continue. This has given us an opportunity to create a resource to investigate the metabolic profiles of available Rhodophyte assemblies using an in silico analysis. By leveraging both annotated and unannotated assemblies across both genomes and transcriptomes, we were able to represent assemblies from across all seven Rhodophyte classes and included assemblies without published annotations, such as *Kappaphycus alvarezii* and *Asparagopsis taxiformis*. This covers a much wider range of red algae than datasets from current publicly available sources, such as Phycocosm [34], which currently houses 18 annotated genomes and no transcriptomes. Furthermore, this allowed for a close look into the diverging metabolic profiles of Cyanidiophyceae between two represented orders, the Galdieriales and Cyanidioschyzonales. 

However, despite the inclusion of transcriptome assemblies and unannotated assemblies, Rhodophyte assemblies are heavily skewed towards only a few classes, most drastically Florideophyceae and Cyanidiophyceae. Within Florideophyceae, assemblies were exclusively sourced from the subclass Rhodymeniophycidae. While Rhodymeniophycidae is the largest subclass of Florideophyceae, with 74% of species listed [1], other subclasses hold ecological significance. The subclass Nemaliophycidae, which holds 12% of Florideophyte species, is the most biologically diverse [35], while the Corallinophycidae, which contains 13% of Florideophyte species, includes a great number of coralline algae, which were not represented in this dataset. 

Furthermore, the substantial differences between the genome and transcriptome protein sequences, particularly the 1KP proteins, prevented a more in-depth analysis of the duplication of metabolically involved genes, while the lower quality of several genome and transcriptome sequences led to the underrepresentation of several classes, particularly Bangiophyceae and Rhodellophyceae. While the MMETSP transcriptomes mostly belonged to either Porphyridiophyceae or the remaining unrepresented classes, only ten were available, and only 5 of those made the standard used. Of the 1KP transcriptomes, only 7 were not Florideophytes, and only 4 of those met the standard used. Thus, for further in-depth analyses to be performed, more assemblies will be required from these Rhodophyte classes. Already, there have been some red algal genomes published on NCBI that could expand the selection, such as the Stylonematophyte *Rhodosorus marinus* and the coralline Florideophyte *Porolithon onkodes*, although it is a first draft genome. Further expansion of the available genome assemblies into the Bangiophytes would allow for a much deeper investigation of red seaweed metabolic profiles.

### 3.2. Repeat Elements Support Evidence of Genome Expansion

The proportion of repeats in Rhodophyte genomes was highly variable but overall showed results consistent with previous studies, with macroalgal assemblies having much higher repeat content than microalgae [26,36], which is indicative of historical genome expansion. Bangiophytes showed higher repeat content than previously reported; however, this may be attributed to different methods in which repeats were accepted in the repeat library, and the small sample size prevented more significant investigation into Bangiophyte repeat content. However, the Bangiophytes showed a noticeably large proportion of tandem repeats, as was previously reported [36]. The Cyanidiophytes, particularly the Galdieriales, showed far more unclassified repeats, while the *C. merolae* assemblies also showed large portions of DNA transposons. However, this will require further investigation, as one *C. merolae* and the *Cyanidiococcus yangmingshanensis* assemblies showed repeat compositions closer to the *Galdieria* assemblies. 

### 3.3. Core Metabolic Pathways Were Conserved between Rhodophyte Classes, but Secondary Metabolic Pathways Were Not Well Represented

Overall, metabolic pathway completion was conserved across Rhodophytes, with core metabolic pathways well represented across all organisms. However, variation was observed in the 1KP transcriptomes. This was most likely due to differences in the sequencing and assembly pipelines used by the 1KP assemblies, which was seen in the lower protein count and mean protein length. The high conservation of metabolic pathways between organisms is also representative of the range of pathways on the KEGG database. The majority of pathways on the KEGG database are core metabolism pathways, such as carbohydrate and amino acid biosynthesis pathways or core energy production pathways. Secondary metabolic pathways on KEGG are largely constrained either to core pathways that produce widely used metabolic precursors found in many metabolites, such as the mevalonate and non-mevalonate pathways, or pathways that produce metabolites that are largely found in bacteria or fungi. Algal specialised metabolites are currently not well represented, as many of their pathways have not been experimentally confirmed. For example, the Rhodophyte metabolite carrageenan is an economically important metabolite produced via the farming of the red seaweed *K. alvarezii* and yet does not have a published pathway with experimental verification. A carrageenan biosynthesis pathway was proposed by Ficko-Blean et al. (2015) [37] and further explored by Lipinska et al. (2020) [38], but it does not yet have any orthologues on the KEGG database.

### 3.4. Sterol Biosynthesis Pathways Were Annotated as Incomplete despite Evidence of Their Production in Red Algae

Previous studies have shown the presence of the bioactive phytosterols sitosterol and campesterol in red algae [8,9,39], as well as other phytosterols such as stigmasterol [8,9,32], ergosterol, and fucosterol in certain Rhodophytes. Campesterol and sitosterol, particularly β-sitosterol, are common bioactive phytosterols. As nutraceuticals, campesterol and β-sitosterol help lower cholesterol absorption in the body while also having anti-cancer activity [40,41,42,43]. Campesterol and sitosterol are the only two phytosterols with pathway modules available on KEGG, but due to the conserved structure of phytosterols, parts of this pathway are likely involved in the biosynthesis of other phytosterols. Furthermore, derivatives of these phytosterols, both natural and synthetic, can have medicinal use, for example, β-sitosterol-3-O-glucoside and heparin-β-sitosterol [42]. In this study, we discuss the incomplete annotation of sterol biosynthetic pathways in red algae despite in vitro evidence of their biosynthesis.

A 2015 study by Santos et al. [8] showed that red algal samples lacked observable amounts of 24-methylenecholesterol, while a 2000 study by M.G Tasende [9] showed that C. crispus lacked 24-methylenocholesterol in its gametophytes but not sporophytes. This corresponded to the absence of the 24-C-methyltransferase orthologue in red algae. Conversely, Chlorophytes have been shown to produce 24-methylenecholesterol [8], and the Chlorophytes in the outgroup showed the sterol 24-C-methyltransferase orthologue. However, given that 24-methylenecholesterol is a precursor to campesterol, either this enzyme or a similar enzyme should be present in red algae in order to facilitate this pathway unless a novel algal phytosterol biosynthesis pathway exists.

Furthermore, in addition to the phytosterol biosynthesis module, most assemblies also showed an incomplete cholesterol biosynthesis pathway, in particular with two orthologues consistently missing across most red seaweeds, while the ergocalciferol pathway was incomplete across almost all algae barring *P. purpureum*, despite the fact that the majority of the sterol content of red algae is cholesterol [8,9] and that previous studies have shown ergosterol, an ergocalciferol precursor, in red seaweed [39]. This indicates that algae may possess alternative pathways for the biosynthesis of these products or that the orthologues of these enzymes show enough sequence dissimilarity to interfere with their annotation using the current orthologues in the KEGG database. 

This evidence supports the findings of Belcour et al. (2020) [32], which used a new bioinformatic tool to infer a new putative cholesterol biosynthesis pathway in *C. crispus*. This study found that while most of the cholesterol pathway was complete, several enzymes were not annotated, which varied by clade. Amongst almost all assemblies, even the outgroup, the 3-keto steroid reductase, which catalyses the biosynthesis of zymosterol, was not observed in the KEGG annotation, which correlated to Belcour et al.’s findings where zymosterol was not observed in *C. crispus*, as well as evidence from Desmond et al. (2009) [44] which failed to detect C-3 ketoreduction in *C. merolae*, *O. tauri*, and other plants. Likewise, the cholestenol delta-isomerase orthologue K01824 was not observed in non-Cyanidiophyte algae, which supports Belcour et al.’s findings that no delta7/8 isomerase orthologues were found in *C. crispus*, while the C-8 sterol isomerase K09829, which is part of the ergosterol biosynthesis pathway, was found in almost no assemblies. In contrast, the cycloeucalenol cycloisomerase orthologue K08246, which is found in the phytosterol biosynthesis pathway, was found in almost all assemblies.

Interestingly, 7-dehydrocholesterol (K00213) reductase was not observed in the Cyanidiophyte algae, while it was annotated in most other assemblies. Likewise, delta24-sterol reductase (K09828) was not annotated in the Galdieriales, while cholestenol delta-isomerase was, indicating that Cyanidiophytes may possess a sterol biosynthesis pathway that is distinct from other Rhodophytes.

The desaturases involved in sterol biosynthesis also showed distinctions between Rhodophyte classes. The sterol 22-desaturase orthologue K09831, which is unique to the ergosterol biosynthesis pathway, was also absent from the Florideophytes and Bangiophytes while being present in the Cyanidiophytes and in the *Porphyridium* genomes. This is supported by Belcour et al. (2020) [32], which showed no ergosterol production in *C. crispus*, as well as Barone et al. (2020), which showed that *G. sulphuraria* can produce ergosterol [45]. However, the Delta7-sterol 5-desaturate orthologue K00227, which is found in all three pathways, was found across almost all assemblies. Notably, however, the *Asparagopsis* genomes did not show annotations for either orthologue.

### 3.5. Variably Completed Pathways Reveal Orthologue Differences in Key Clades

The distribution of variably completed pathway modules between Rhodophyte classes largely followed the distribution of modules to module classes (e.g., carbohydrate metabolism, amino acid metabolism) but less so to module categories (e.g., central carbohydrate metabolism, pyrimidine metabolism), implying that overall pathway variations were not weighted towards specific module classes, but that specific types of pathways varied between organisms. In terms of amino acids, aromatic amino acids were the most prevalent in the variably completed pathways, in particular, pathways involving phenylalanine and tyrosine biosynthesis. In those pathways, the difference was due to a singular KEGG orthologue K01850 (chorismate mutase) not being predicted in eight of the *G. sulphuraria* genomes or *C. caldarium*, while the remainder of these pathways were similar to other organisms, which may imply a difference in aromatic amino acid pathways in *G. sulphuraria*. Similarly, differences were observed in other pathways, such as the ascorbate biosynthesis pathways M00129 and M00114, the purine and pyrimidine biosynthesis pathways M00959, M00546 and M00046, and the melatonin biosynthesis (animals) pathway M00037, due to consistent patterns of orthologue presence or absence in the Cyanidiophytes.

### 3.6. Differences between Cyanidiophyte Orders Likely Due to Environmental Adaptations

Prior evidence shows that Cyanidiophyceae diverged early from other Rhodophyte classes [46]. As such, several distinct trends were observed in the Cyanidiophytes but not in other Rhodophyte algae, such as the high duplication of transmembrane transporters or in orthologue presence across a variety of pathways. However, divergence was also observed within Cyanidiophyceae. Three orders within Cyanidiophyceae were represented here, including Galdieriales (*Galdieria* spp.), Cyanidiales (*C. caldarium*), and the Cyanidioschyzonales (*C. merolae* and *C. yangmingshanensis*). The Galdieriales and *C. caldarium* often showed consistent differences from the Cyanidioschyonales, such as the larger proportion of DNA transposons in *C. merolae*, the higher proportion of transmembrane transporter duplications in *Galdieria* and *Cyanidium*, and through patterns of orthologue presence or absence. These two clades of organisms, while similar, still have systemic differences, which might explain these differences. While all three are polyextremophiles living in hot, highly acidic environments, *Galdieria* and *Cyanidium* spp. possess a higher tolerance to salt concentration than *C. merolae*, while also possessing a rigid cell wall, which *C. merolae* does not [47], which correlates to the increased duplication of transmembrane transporters in *Galdieria* and *Cyanidium*. 

Previous evidence has also shown that Cyanidiophytes lost genes involved in urea utilisation but that *G. phlregrea* had regained these loci due to horizontal gene transfer [48]. The evidence shown in Figure 3C supports the evidence published by showing the absence of urea-utilising genes in the urea utilisation modules M00546 and M00959, though they showed incomplete pathways both for *G. phlegrea* and *G. sulphuraria*. The recent publication of many Cyanidiophyte genome assemblies shows a potential avenue for further research into extremophilic Rhodophyte evolutionary adaptations to extreme environments. However, some intraspecies variation in these was observed within the *Galdieria* assemblies, while the *Cyanidioschyzon* assemblies were uniform. Given that some of the variable genes correlated with low BUSCO scores in some *Galdieria* assemblies, it is possible that more complete assemblies would show higher rates of presence of these genes. However, this would not contradict the correlation between order and gene presence where there was a difference between *Cyanidioschyzonales* and *Galdieriales*, but would rather increase the correlation values shown.

### 3.7. Duplication of Haem Peroxidases Correspond to Higher Production of Halogenated Compounds

Like the Cyanidiophytes, the Florideophyte algae showed systemic differences to other classes via gene duplications, with haem peroxidases highly duplicated across the class. This reflects the expansion in animal-heme peroxidases reported by Collén et al. (2013) [16], which supports their assertion that it is an adaptation in red algae to life in marine environments where halogenated compounds are common and utilised in cellular defence and secondary metabolism. These adaptations have also been observed in brown algae, which also utilise halogenated metabolites in a similar manner [49]. It also suggests that this large expansion of heme-peroxidases was most pronounced in the Florideophytes and not observed in other Rhodophyte classes.

Haem peroxidases form a wide range of enzymes that are involved in oxidation reactions utilising hydrogen peroxide as an oxygen acceptor and are involved in the biosynthesis of various metabolites, including secondary metabolites in plants and algae, as well as other cellular functions such as cell signalling and defence [50,51,52]. Further analysis of the annotations of these enzymes indicates that this gene duplication may have resulted in the subfunctionalisation of these genes. The majority of heme peroxidases inferred to be duplicated were annotated as animal heme peroxidases, which is an old denomination yet still present in some annotation databases and is associated with the oxidation of halides [52], while others were annotated as Peroxinectin, a multifunctional molecule with adhesive and peroxidase activity [53]. Both belong to the peroxidase-cyclooxygenase superfamily [52,53]. *G. lemaneiformis* and *G. chorda* also had several genes annotated as eosinophil peroxidase, a heme peroxidase involved in the oxidation of bromide ions [54]. However, further research would have to be performed to fully explore how gene duplication was involved in the subfunctionalisation of heme peroxidases in Rhodophytes. Given that Florideophyte algae are known for their production of brominated natural products that rely on the production of hydrogen peroxide in their biosynthesis [14,55], the duplication of halide-associated haem peroxidases highlights the evolution of towards the production of halogenated secondary metabolites in red seaweeds, and in particular towards the utilisation of brominated compounds in Florideophyte seaweeds.

### 3.8. Future Rhodophyte Innovation Needs Greater Publication of High Quality, Annotated Assemblies and Greater Metabolomic Integration

This study used a broad bioinformatic approach to compare metabolic trends across Rhodophyte assemblies in both genomes and transcriptomes. However, the lack of annotated genome assemblies and the relatively low quality of several of these assemblies hampered effective analysis. It has been reported that in some metrics, algal assembly quality has gone down, such as via the increased publication of fragmented genome assemblies without scaffolding [19]. Rhodophyte assemblies, in particular, also do not have a dedicated database for assembly quality testing with BUSCO and typically show lower scores on average compared to green algae [19,56].

This study has also shown incomplete biosynthetic pathway annotations of key metabolites, including sterols and nucleotide biosynthesis pathways using the KEGG database, despite in vitro evidence of their production in red algae, indicating that novel genes or pathways are involved in these biosynthesis genes. This also contrasts with other data sources, such as the recent description of an algal cholesterol biosynthesis pathway from *C. crispus* [32]. Thus, this study suggests further research to develop our understanding of Rhodophytes and the development of bioinformatics tools and datasets designed for use with Rhodophyte assemblies, such as a BUSCO database, and for further modelling and biochemical analysis of algal metabolic pathways, including those which are otherwise well known in other organisms, and their inclusion into public databases used for functional annotation of assemblies.

## 4. Materials and Methods

This study was conducted using a custom bioinformatics pipeline (Figure 5) utilising open-source programs and the OmicsBox bioinformatics platform, utilising publicly available data. Genome and transcriptome assemblies were sources from the NCBI assembly database and from individual study datasets. Red algal assemblies were searched for in the NCBI assembly database using the terms “Rhodophyta” and “red algae”. Datasets linked to assemblies available on NCBI were also searched for annotated assemblies. 10 Cyanidiophyte assemblies were sourced from the project dataset of the Rossoni et al. (2019) study [2,20], which included functional annotations. A draft *Asparagopsis taxiformis* genome was sourced from the project data of the relevant study [25]. Red algal transcriptomes were sourced from the MMETSP project [21], which had been reassembled and cleaned by [57], as well as from the One Thousand Plant Transcriptomes (1KP) [22] project database [23]. To form a phylogenetic outgroup, assemblies from green algae (Chlorophyta) and Glaucophyta were downloaded from the NCBI database. The Glaucophyte and a small subset were selected for functional annotation, while the remainder of the outgroup was used only for general statistical comparisons (Appendix A). Assembly acquisition was completed in April 2022. Taxonomic data were retrieved from AlgaeBase [1].

Initial assembly quality was assessed using BUSCO [58] on genome or transcriptome mode as relevant against the Eukaryota_odb10 dataset. AUGUSTUS [59] was used as the gene predictor in BUSCO genome runs, and MetaEUK [60] for transcriptome runs, as transcriptome mode does not support AUGUSTUS. AUGUSTUS results from BUSCO were used as training data for downstream gene prediction. As transcriptome mode did not generate AUGUSTUS training data, transcriptome assemblies were also analysed with BUSCO on genome mode to generate training data. Galdieria was used as the species for red algal BUSCO analyses, and green algae were run using *Volvox carteri, Chlamydomonas reinhardtii* and *Galdieria*.

Protein annotation of the genome assemblies required two steps: repeat masking and protein prediction. To identify the repeat content of each genome assembly, the programs RepeatModeler v2.0.2a [61] and RepeatMasker v4.1.2 [62] were used. RepeatModeler was used to create custom repeat libraries for each genome assembly, and RepeatMasker was used to identify the repeats of all genome assemblies and soft-mask the assemblies without prior protein annotations. Transcriptome assemblies were not investigated for repeat content. Masked assemblies from unannotated assemblies were used in gene prediction, while repeat masking in pre-annotated genomes was only used for comparison. Identified repeats were classified as either interspersed repeats, which included categories for retroelements and DNA transposons, or other repeats, such as satellites, low complexity repeats, and small RNA repeats. The protein prediction program AUGUSTUS v3.4.0 [59] was used to predict the protein-coding genes of each unannotated assembly. The training data generated by the initial BUSCO tests was used to train AUGUSTUS for each prediction by using the training data generated by the AUGUSTUS run in the BUSCO test as the species for the protein prediction. Final protein annotation quality was then assessed using BUSCO on protein mode against Eukaryota_odb10. To provide a comparison for final BUSCO quality, 41 green algal assemblies with existing protein annotations available were also assessed using BUSCO on protein mode.

Functional annotation was performed using the BLAST2GO suite [63] via the OmicsBox v2 platform [64]. Unannotated genomes and the MMETSP transcriptomes were BLASTed against the BLAST NR database. Species with multiple assemblies were first clustered with CD-HIT v4.8.1 [65] with alignment coverage for both longer and shorter sequences set to 0.9 and set to similar to the most similar cluster, and the resulting clustered sequence BLASTed against NR, and the clustered sequences used as a custom BLAST database for the original sequences. Functional annotation was then performed on all individuals using InterPro [66], GO annotation [67,68], eggNOG [69], and KEGG [70,71,72]. The genome assemblies from the Rossoni et al. (2019) [20] dataset had their functional annotations reannotated in order to match the other genomes, but protein and BLAST annotations were retained. Where KEGG annotations were not predicted but likely to exist, genes were taken from the KEGG database and searched for in the organisms either through their original databases, or against the resource created for this study, using BLASTp, BLASTn, or tBLASTn or BLAT [73] as appropriate.

Metabolic reconstruction was performed using the Reconstruct function of the KEGG Mapper tool [72,74,75], using the KEGG and eggNOG annotation results, and was complemented with a search of each assembly for the KOs of each module. Each metabolic pathway is represented by one or more KEGG modules, which are divided into blocks composed of one or more orthologues. Each block represents an enzymatic step in a reaction. A block can also be complex, made up of multiple KOs, where every KO has to be present for the block to be recognised as present. KEGG Reconstruct categorizes pathways as either being complete, having one or two blocks missing, or incomplete (at least one block present, but more than two missing). Modules with no blocks present were marked as absent. These were then represented using a numeric scale ranging from one (complete) to five (absent). Furthermore, some modules contain gene complexes in which multiple KO steps form a single block, and all must be present for it to be complete. In cases where partial blocks due to complexes were detected, their completion was graded manually using how many KO “steps” were missing, with each step being treated as equivalent to a block (so a complex with one KO missing would be treated as having one block missing, and so on). Metabolic pathways between Rhodophytes were further compared using rates of completion of specific pathways across Rhodophyte classes. To identify modules with different levels of completion, the proportions of complete pathways for each class were calculated, with variance measured by taking the standard deviation of these values (Figure 3A). As the transcriptomes had significant differences in total pathway completion compared to the genomes, they were grouped separately by source. Only pathways that were loosely complete in at least one red algal assembly were used.

Phylogenomic relationships were inferred using Orthofinder v.2.5.4 [24,27,28]. Approximate phylogenies of all samples were inferred using Diamond as the search engine [76] and DendroBLAST as the inference program [77], while more accurate phylogenies of genomes were inferred using MAFFT [78,79] and FastTree [80] for multiple sequence alignment and tree inference, respectively. MSA trees were validated by constructing and validating trees with RAxML-NG. Model selection was performed on multiple sequence alignments using the MEGA11 software v11.0.13 [81]. Phylogenomic relationships in the Cyanidiophyes were supported by additional phylogeny inferred using BuscoPhylo [29] on both protein and genome mode, using *P. purpureum* UTEX161 as an outgroup. Metabolomic data was linked to phylogenetic tree nodes using the gene duplication data outputted from OrthoFinder and functional annotations for each gene retrieved. Gene duplication analysis and requisite phylogenetic inference were only performed on genome assemblies due to systematic differences between the genome and transcriptome assemblies. Phylogenetic trees were visualised with iTOL [82].

Statistical analysis was performed with the Microsoft Excel Data Analysis Toolpak. Assembly statistics, including BUSCO completion rates, protein counts, mean protein length and repeat content, as well as pathway completion rates, were analysed using single-factor ANOVA tests for multiple categories or 2-tailed t-tests assuming unequal variances for comparisons between two groups. The significance threshold for all statistical tests was set at 0.05. Statistical analysis for functional annotations and pathway analysis was only performed on assemblies that met the BUSCO quality threshold of at least 35% complete and less than 50% missing.

## 5. Conclusions

The red algae area is a major focus of current scientific research, with Rhodophyte metabolites and biosynthesis pathways being a particular focus. The recent uptick in the Rhodophyte genome and transcriptome assembly publication has allowed for a large-scale analysis of metabolic pathways across red algae. By integrating functional annotation with metabolic reconstruction and phylogenomic analysis, this study was able to use a resource to explore the metabolic pathways and gene duplication of functional components associated with medicinally active compounds. Metabolic reconstruction using KEGG annotation showed incomplete annotation of sterol biosynthesis pathways despite in vitro evidence of sterol biosynthesis in Rhodophytes, possibly due to novel gene orthologues not described by the KEGG database. This supports prior evidence of alternate sterol biosynthesis in red algae. Further investigation into metabolic trends also highlighted high rates of gene duplication of haem peroxidases responsible for halide oxidation in Florideophyte algae but less or no duplication of these genes in other phylogenetic nodes, which corresponds to their high rate of production of halogenated secondary metabolites.

## Figures and Tables

**Figure 1 marinedrugs-22-00003-f001:**
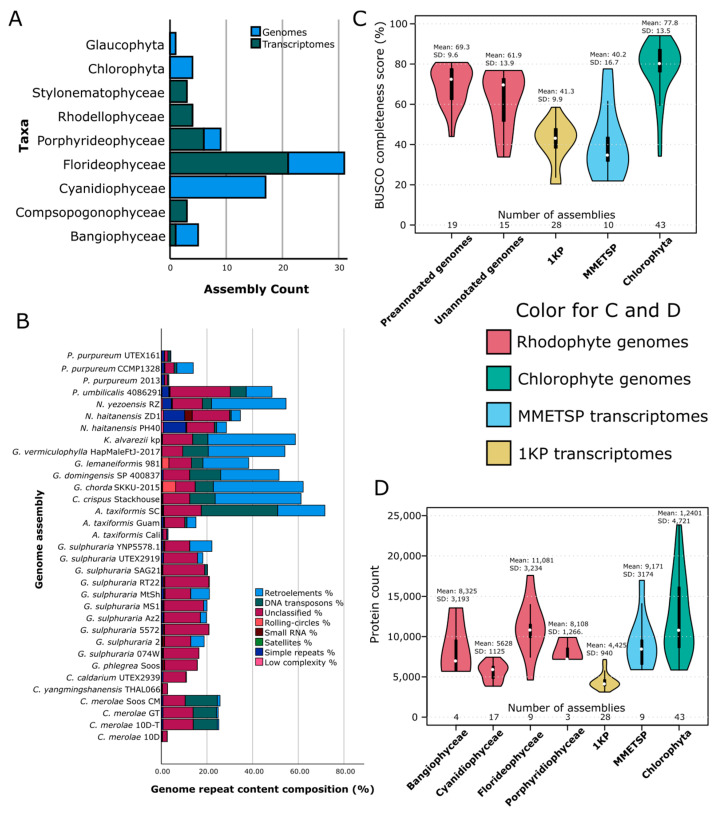
Overall assembly statistics for Rhodophyte assemblies. (**A**). Total counts of genome and transcriptome assemblies were used in this study. Chlorophyte assemblies that only had their assembly statistics measured were not included in this count. (**B**) Repeat content of Rhodophyte genome assemblies as predicted by RepeatModeler and RepeatMasker. Repeat types were collated into overarching categories as described by RepeatMasker. (**C**) Violin plot of the BUSCO scores of the final protein annotations for each assembly. Assemblies with downloadable protein annotations were classified as pre-annotated, while those without prior annotations were classified as unannotated. (**D**) Protein counts of Rhodophyte genome assemblies by class, transcriptome assemblies by source, and an outgroup. Chlorophyte protein counts were taken from pre-annotated genome assemblies from the NCBI Assembly database.

**Figure 2 marinedrugs-22-00003-f002:**
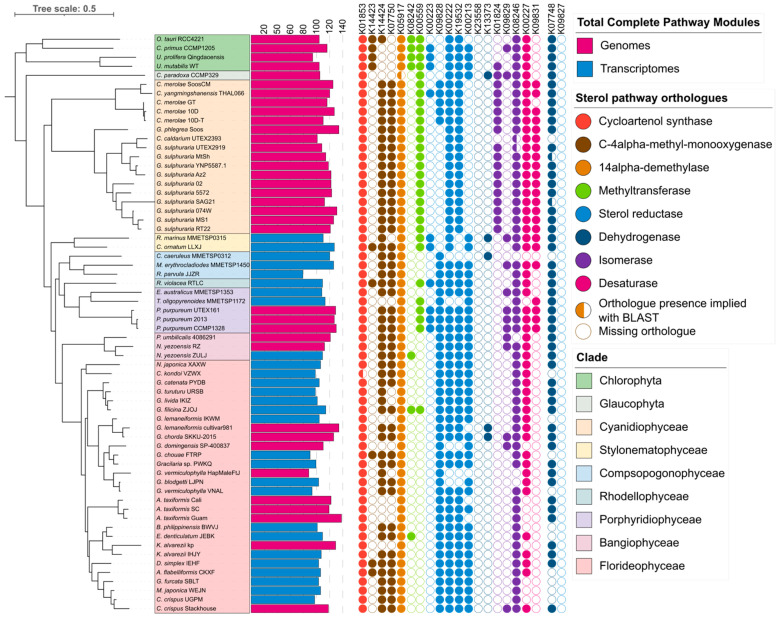
Comparison of complete pathways across Rhodophyte assemblies and predicted annotations of orthologues belonging to sterol biosynthesis pathways. An approximate phylogenetic tree was inferred via OrthoFinder using DendroBLAST. The number of complete pathways includes both fully complete pathways and those with one block missing or one gene missing if the gene is part of a gene complex. Enzymes shown belong to the biosynthesis pathways for phytosterol (sitosterol and campesterol), cholesterol (M00101), and ergocalciferol/ergosterol (M00102). Coloured circles represent orthologues marked as present by KEGG, semi-circles represent orthologues that were not inferred by KEGG but had BLAST evidence supporting their presence, while empty circles represent orthologues not predicted by KEGG. The orthologues involved in C4-demethylation are shown at the end of the table.

**Figure 3 marinedrugs-22-00003-f003:**
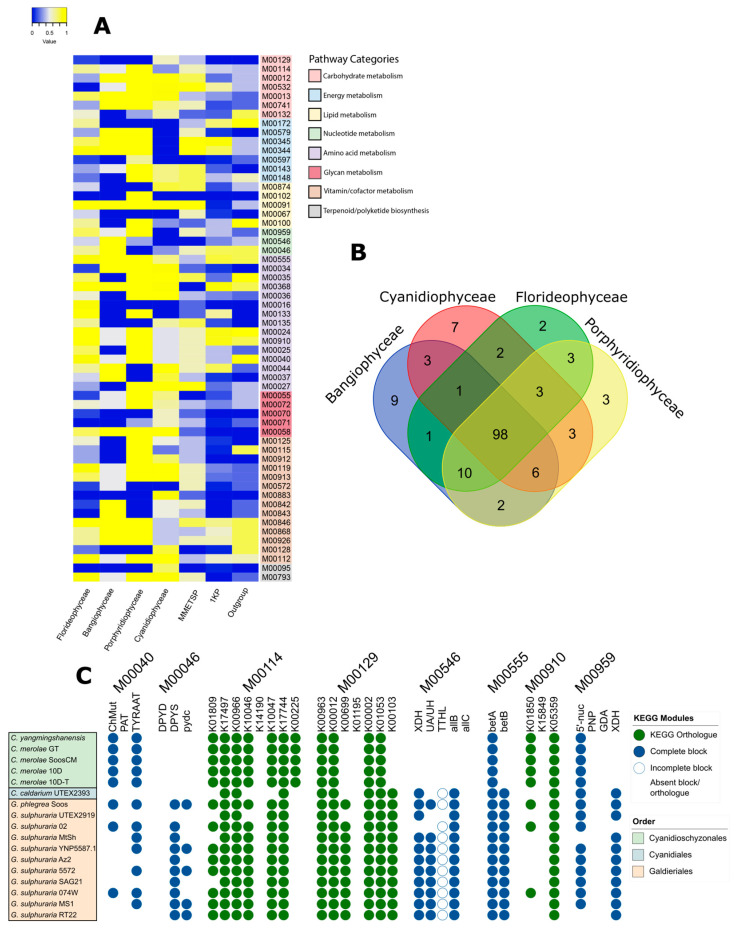
Comparison of metabolic pathways with highly variable rates of completion between Rhodophyte classes. (**A**) KEGG pathway modules with the most variable rates of completion by red algal class or transcriptome source. Pathways with a standard deviation of the population of 0.2 or higher were selected and grouped by pathway category. Rates of completion range from 0 (none complete) to 1 (all complete). Pathways with one block missing were counted as loosely complete and included in this calculation. (**B**) Venn diagram of KEGG modules for each class. Only complete or loosely complete modules found in 50% of genome assemblies for each class were counted. (**C**) Completion rates of 8 modules from (**A**) showed distinct differences between Cyanidiophytes of the orders Galdieriales and Cyanidioschyzonales. *C. caldarium* of the Cyanidiales was most similar to the Galdieriales. Modules where each orthologue forms a block are shown in green, while modules with blocks representing multiple orthologues are shown in blue. Blank spaces without circles represent absent orthologues/blocks.

**Figure 4 marinedrugs-22-00003-f004:**
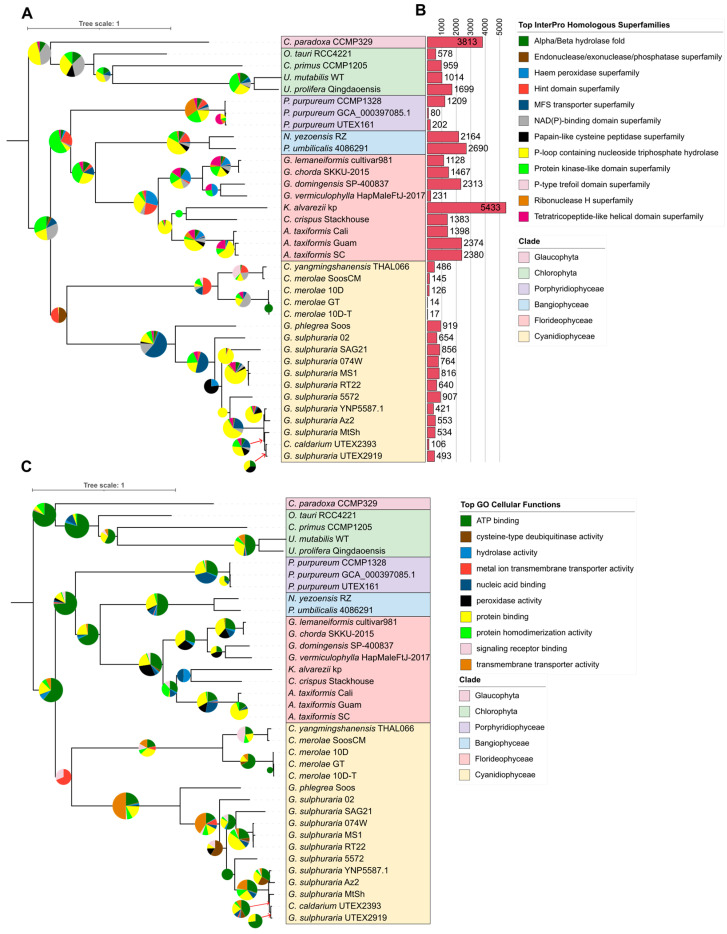
Gene duplication trees of 36 annotated genomes. Phylogeny was inferred using OrthoFinder in MSA mode, with duplications of major nodes integrated with (**A**) InterPro Homologous Superfamily annotations and (**B**) GO Cellular Functions. Both figures utilise the same phylogenetic tree, which was rooted at the midpoint in iTOL. Total duplications for each assembly node are shown in (**C**). Categories shown include the most abundant annotation for each node, barring those in small quantities that only appear on one node and do not represent total proportions of annotations. Only duplications with at least 50% support are shown. Pie graph radii are proportional to the log_10_ of the total number of duplications for each node.

**Figure 5 marinedrugs-22-00003-f005:**
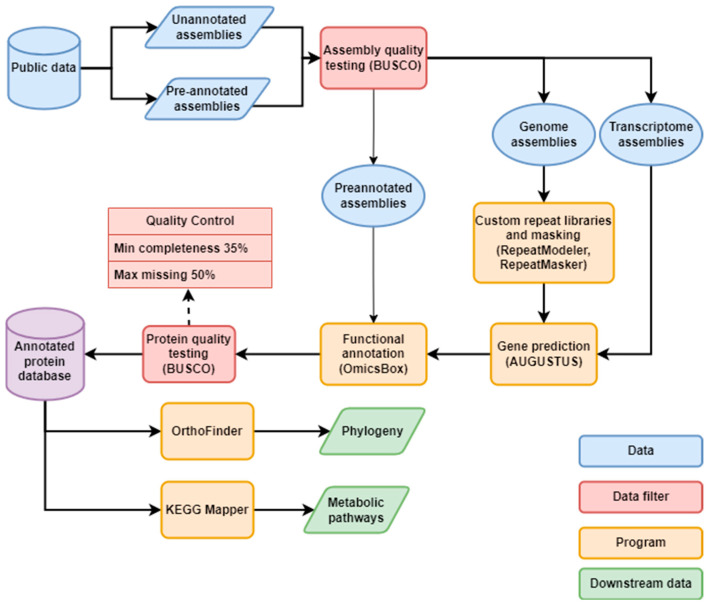
The bioinformatics pipeline used in this study.

**Table 1 marinedrugs-22-00003-t001:** Correlation of the presence or absence of KEGG orthologues and blocks against taxonomic order. Positive correlation indicates presence in Galdieria and absence in the Cyanidioschyzonales, while negative correlation indicates the opposite.

Module	Orthologue	Function	Correlation
M00040	CHM	Chorismate mtuate	−0.68
M00046	DPYS	dihydropyriminidinase	0.77
M00114	K00225	L-galactono-1,4-lactone dehydorogenase	−1.00
M00129	K00699	glucuronosyltransferase	0.77
M00129	K00103	L-gulonolactone oxidase	1.00
M00546	XDH	xanthine dehydrogenase	0.87
M00546	UA/UH	urate hydroxylase	0.68
M00546	TTHL	5-hydroxyisourate hydrolase	0.87
M00546	allB	allantoinase	1.00
M00555	K14085	betaine-aldehyde dehydrogenase	1.00
M00910	K01850	chorismate mutase	−0.68
M00959	XDH	chorismate mutase	0.87

## Data Availability

The genome and transcriptome sequence data presented in this study, as well as BUSCO, RepeatMasker and OrthoFinder results, are openly available in the University of the Sunshine Coast institutional repository at https://doi.org/10.25907/00782, accessed 16 November 2023.

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
