# Peer review of "Identification of Incomplete Annotations of Biosynthesis Pathways in Rhodophytes Using a Multi-Omics Approach"

_marinedrugs, 2023, doi:10.3390/md22010003_

Round 1

Reviewer 1 Report

Comments and Suggestions for Authors

The manuscript reports a reanalysis of existing rhodophyte genomes and transcriptomes and lineage-specific variations in major metabolic pathways. Given the large dataset and many taxa (72 species from all seven classes of red algae) covered by the analysis, the work is of interest and potentially substantial value. I enjoyed reading the manuscript, which is well written in the most part (but see below for specific comments, e.g. methods being described in the result section). I do have a number of comments that need to be addressed to make the paper acceptable and the eventual publication useful for future research. With that, I recommend a moderate (somewhat extensive) level of revision.

Specific comments are given below.

Title is a bit misleading because it gives the impression that algae are not able to synthesize sterols. Rather the work shows lineage-specific variability of metabolism and potentially unique sterol biosynthesis pathways in red algae. Besides, although haem peroxidases duplication is highlighted in the title, results and discussion do not really do justice. I suggest that the title be revised to reflect the real findings and conclusions.

The Results section contains a high amount of method information, which should be moved to the Methods section.

Figure 1D: The legend should explain what the numbers above the X-axis depict. Also, I would suggest that the mean plus/minus SD of protein count be shown, perhaps on top of the boxes for each group. Numbers are more useful for other researchers than plots, while plots make it easy to see trends. A supplementary table showing the protein count for each species should be presented.

Figure 1C carries potentially useful information for future use of the existing genome/transcriptome data, but to make it really useful, a supplementary table is needed to show detailed metadata of species, strain, genome project ID (if available) and source (NCBI, or elsewhere), and the BUSCO completeness score.

Figure A3 is potentially very useful, but the pie charts are too small to see the detail. It is important to either make them bigger, with some placed outside the node but pointed to the node with an arrow, and/or supplement this with a supplementary table showing the percentage values for each ortholog.

There is a concern about the high fraction of pathways that are incomplete: as reported in section 2.2, a total of 410 pathway modules were analyzed, of which 384 were pathways modules and 26 were complex modules…in total 37 modules were complete in all assemblies though only 3 when using the strict standard, while 74 were absent in all assemblies. This indicates that either red algae pathways are distinct compared to those in KEGG or genome/transcriptome data are incomplete, instead of that red algae do not have functional pathway to carry out so many metabolic processes. I understand that the authors gave this due discussion, but the tone before the Discussion section gives strong implications that these pathways are truly incomplete.

A closer look at the pathway that this manuscript focuses on. “All individuals showed an incomplete phytosterol pathway module; however, all assemblies reported at least four of the KEGG orthologues required, with a median of eight orthologues present across all assemblies…… Similar to the phytosterol module, the cholesterol and ergosterol biosynthesis modules, M00101 and M00102 respectively, were largely incomplete across the red algal assemblies.” Then, “This contradicts in vitro evidence of sterol production in red algae, indicating that red algae may have novel genes involved in the biosynthesis of sterols that have not been identified in the KEGG database.” The last statement, which needs a reference to back up, echoes the question I posed above. The authors need to assess more closely and come up with a conclusion whether this last statement applies to all the apparently incomplete pathways.

Figure 3C: the legend on the right explains what the green fill, blue fill, and open circle depict, but does not explain what the blank spots (with a circle at all) mean. Besides, it is striking for intraspecific populations of Galdaria sulphuraria to vary so remarkably in many of the modules. To me, this is evidence that there are extensive assembly and/or annotation inconsistencies. If the authors do not think assembly/annotation inconsistencies might be the reason, please provide an argument about it.  

Author Response

1.

Reviewer: Title is a bit misleading because it gives the impression that algae are not able to synthesize sterols. Rather the work shows lineage-specific variability of metabolism and potentially unique sterol biosynthesis pathways in red algae. Besides, although haem peroxidases duplication is highlighted in the title, results and discussion do not really do justice. I suggest that the title be revised to reflect the real findings and conclusions.

Response: We have adjusted the title to more clearly insinuate that it is the annotations that are incomplete, rather than the pathways themselves. We have put more focus in the title on our evidence supporting alternate biosynthesis pathways within Rhodophytes. Thus, we suggest an alternate title:

Identification of incomplete annotations of biosynthesis pathways in Rhodophytes using a multi-omics approach

2.

Reviewer: The Results section contains a high amount of method information, which should be moved to the Methods section.

Response: We have moved method information from the results into the methods. Where instances of duplication between methods and results were found, they were deleted from the results. These changes were extensive, so we will list a short summary of changes:

  • Moved section regarding collection of outgroup genomes to methods, reworded paragraph to accommodate changes, instead focusing on the number retrieved.
  • Taxonomic retrieval from AlgaeBase moved to methods.
  • Moved details of repeat prediction and masking to results
  • Moved details of AUGUSTUS annotation to methods.
  • Shortened metabolic reconstruction introduction (2.2) to read more as results than methods.
  • Moved section on comparison of pathway completion (2.2.2) between classes to methods.
  • Altered results section on gene duplication analysis to move methods statements to methods.

We have left mentions of total numbers of assemblies retrieved in the results, as we wrote it as the search being the methods, and the total number retrieved the results. However, we can move all mention of total numbers of assemblies retrieved to methods, if requested.

3.

Reviewer: Figure 1D: The legend should explain what the numbers above the X-axis depict. Also, I would suggest that the mean plus/minus SD of protein count be shown, perhaps on top of the boxes for each group. Numbers are more useful for other researchers than plots, while plots make it easy to see trends. A supplementary table showing the protein count for each species should be presented.

Response: Figure 1C and 1D are now given a header to show that the numbers at the bottom indicate number of assemblies. Mean and SD values are shown as labels above violins. Protein counts have also been added to Supplementary Table 1.

4.

Reviewer: Figure 1C carries potentially useful information for future use of the existing genome/transcriptome data, but to make it really useful, a supplementary table is needed to show detailed metadata of species, strain, genome project ID (if available) and source (NCBI, or elsewhere), and the BUSCO completeness score.

Response: We have added columns to supplementary table 1 containing strain/intraspecific ID, Bioproject, and year of publication, as well as total protein counts and full BUSCO scores.

5.

Reviewer: Figure A3 is potentially very useful, but the pie charts are too small to see the detail. It is important to either make them bigger, with some placed outside the node but pointed to the node with an arrow, and/or supplement this with a supplementary table showing the percentage values for each ortholog.

Response: We have doubled size of pie charts. Additionally, we have created supplementary tables 4 and 5, which contain the raw counts of the annotations used in these pie charts. The data used is also available in Supplementary File S3, though that contains all annotations.

6.

Reviewer: There is a concern about the high fraction of pathways that are incomplete: as reported in section 2.2, a total of 410 pathway modules were analyzed, of which 384 were pathways modules and 26 were complex modules…in total 37 modules were complete in all assemblies though only 3 when using the strict standard, while 74 were absent in all assemblies. This indicates that either red algae pathways are distinct compared to those in KEGG or genome/transcriptome data are incomplete, instead of that red algae do not have functional pathway to carry out so many metabolic processes. I understand that the authors gave this due discussion, but the tone before the Discussion section gives strong implications that these pathways are truly incomplete.

Response: The high fraction of incomplete pathways is due to the broad nature of the KEGG database, where many of these pathways are not applicable to red algae. A large number of these pathways contain bacterial and fungal metabolic pathways, which are not found in plants and algae, whereas others are alternatives to pathways found. We left mention of these pathways in for the sake of completion and comparison, but could be removed, if requested. We have also added the statement (new statement in red): “In total, 37 modules were complete in all assemblies, though only 3 when using the strict standard, while 74 pathways were absent in all assemblies, largely representing bacterial and fungal metabolite biosynthesis pathways.”

7.

Reviewer: A closer look at the pathway that this manuscript focuses on. “All individuals showed an incomplete phytosterol pathway module; however, all assemblies reported at least four of the KEGG orthologues required, with a median of eight orthologues present across all assemblies…… Similar to the phytosterol module, the cholesterol and ergosterol biosynthesis modules, M00101 and M00102 respectively, were largely incomplete across the red algal assemblies.” Then, “This contradicts in vitro evidence of sterol production in red algae, indicating that red algae may have novel genes involved in the biosynthesis of sterols that have not been identified in the KEGG database.” The last statement, which needs a reference to back up, echoes the question I posed above. The authors need to assess more closely and come up with a conclusion whether this last statement applies to all the apparently incomplete pathways.

Response: We have added a reference to a study showing alternative steroid genes in diatoms, as well as a new cholesterol pathway described in Chondrus crispus using in silico analysis by Belcour et al 2020. The section is as follows:

Evidence of alternative sterol biosynthesis genes have been found in other organisms. For example, alternative squalene epoxidase genes, which is key to steroid production in eukaryotes, have been shown to have alternative genes not previously identified in di-atoms and Cryptophyceae [36]. Furthermore, Belcour et al 2020 [37] describes an alternate cholesterol biosynthesis pathway for Chondrus crispus predicted using in silico analysis that differs from the pathway used by the KEGG database.

8.

Reviewer: Figure 3C: the legend on the right explains what the green fill, blue fill, and open circle depict, but does not explain what the blank spots (with a circle at all) mean. Besides, it is striking for intraspecific populations of Galdaria sulphuraria to vary so remarkably in many of the modules. To me, this is evidence that there are extensive assembly and/or annotation inconsistencies. If the authors do not think assembly/annotation inconsistencies might be the reason, please provide an argument about it.

Response:

Point A) We have added a blank spot legend icon for Figure 3C, which shows absent orthologues/blocks.

Point B) We performed a correlation analysis on the variable genes in question, which shows that the variable genes in the Galdieria assemblies do correlate with a lower BUSCO score. However, the main point of Figure 3C is the striking differences between the Galdieria and Cyanidioschyzon assemblies, where one order has specific genes present while the other has them absent. The variable genes between the Galdierias do not interfere with this point. We do not believe that these issues were due to errors with the functional KEGG annotation, as the same process was used for all organisms. It is possible that sequencing or assembly errors are the cause of this. However, this was an analysis of publicly available genome resources, and all bar one of the Galdierias were already published with protein annotations available, and thus this variation reflects more on the assemblies than the organisms themselves.

Thus, we added the following section in the results:

Correlation analysis showed specific gene orthologue or block presence correlated with taxonomic order across all eight pathways, though most genes and blocks were con-served (Table 1). Intraspecific variation was observed within the Galdierias, whereas the Cyanidioschyzonales showed more uniformity. This variation correlated with BUSCO scores in the Galdieria assemblies and in Cyanidium, indicating that these orthologues may be more widespread than predicted with KEGG annotation; however, this does not interfere with of the correlation analyses for taxonomic order as no BUSCO quality correlation was observed within the Cyanidioschyzonales.”

And in the discussion (3.6):

However, some intraspecies variation in these was observed within the Galdieria assemblies, while the Cyanidioschyzon assemblies were uniform. Given that some of the variable genes correlated with low BUSCO score in the some Galdieria assemblies, it is possible that more complete assemblies would show higher rates of presence of these genes. However, this would not contradict the correlation between order and gene presence where there was a difference between Cyanidioschyzonales and Galdieriales, but would rather increase the correlation values shown.”

Reviewer 2 Report

Comments and Suggestions for Authors

Major points: 

The novelty of this paper is not obvious, because the two main conclusions from the title are replicating findings already reported in previous papers that are not cited. 

An explicit model for sterol biosynthesis pathway in red algae has already been published (Belcour et al., 2020; https://doi.org/10.1016/j.isci.2020.100849). This paper already discusses the partial conservation of the sterol biosynthesis pathway, and has already been integrated in the Metacyc database (https://metacyc.org/META/NEW-IMAGE?object=PWY-8191). So the author should discuss how their findings compare with that analysis. By the way this also points toward the fact that kegg is perhaps not the best starting database to reannotate those red algal genomes. 

Similarly, expansion of animal-like peroxidases has already been reported in the genome of Chondrus crispus (page 5 of Collén et al., 2013, https://doi.org/10.1073/pnas.1221259110), so this paper should also be cited and discussed. 

Minor points

lines 49-50:  "however red algae have been shown to possess phytosterols such as sitosterols, campesterol, and fucosterol"

-> although in line with the cited reference, this seems to be a bold oversimplification. According to other authors, like Darnet et al, 2021 (doi:10.3389/fpls.2021.665206), there is variation in the stereochemistry of methyl groups on the lateral side chain, so it would be more prudent to tell that "red algae have been shown to posses small amounts of C24 methyl- and ethylsterols."

Lines 197: I am not sure it is meaningfull to separate phytosterol biosynthetic enzymes from cholesterol/ergosterol ones. Probably there is a strong overlap. For example the identical conservation pattern of K011853 an K011852 goes with the fact that lanosterol synthase and cycloartenol synthase are actually corresponding to the same orthogroup, lanosterol synthase being present only in animals and fungi. So Figure 2 could be simplified by grouping enzymes with similar catalytic activities and probably corresponding to same sequences. 

Lines 256-257: "This contradicts in vitro evidence of sterol production in red algae, indicating that red algae may have novel genes involved in the biosynthesis of sterols that have not been identified in the KEGG database."

-> This has been indeed demonstrated in diatoms (Pollier et al., 2018; https://doi.org/10.1038/s41564-018-0305-5).

Lines 420-421: "A carrageenan biosynthesis pathway has been proposed [43]"

-> a more up do date reference from the same group is Lipinska et al., 2020 (https://doi.org/10.1038/s41598-020-67728-6), where differencial expression of candidate enzymes is discussed. 

Typos: 

lines 20-21: We report that sterol pathways, including campesterol, β-sitosterol, ergocalciferol and cholesterol,

-> including campesterol, β-sitosterol, ergocalciferol and cholesterol biosynthesis pathways

lines 136-137: This corroborates pervious evidence -> previous

Figure 1B: P. umbilicalis should be in italics

Figure 2: Kegg orthologue accession numbers should be indicated somewhere in the legend. 

lines 415: secondary metabolites -> specialized metabolites

Author Response

Major points

1.

Reviewer: The novelty of this paper is not obvious, because the two main conclusions from the title are replicating findings already reported in previous papers that are not cited.

Response: We have altered the title to provide a more accurate description of the work. We have also added in citations and discussions upon those papers mentioned (discussed in subsequent points).

2.

Reviewer: An explicit model for sterol biosynthesis pathway in red algae has already been published (Belcour et al., 2020; https://doi.org/10.1016/j.isci.2020.100849). This paper already discusses the partial conservation of the sterol biosynthesis pathway, and has already been integrated in the Metacyc database (https://metacyc.org/META/NEW-IMAGE?object=PWY-8191). So the author should discuss how their findings compare with that analysis. By the way this also points toward the fact that kegg is perhaps not the best starting database to reannotate those red algal genomes.

Response: We have added a section discussing the pathway described by Belcour et al:

This evidence supports the findings of Belcour et al (2020) [47], which used a new bioinformatic tool to infer a new putative cholesterol biosynthesis pathway in C. crispus. This study found that while most of the cholesterol pathway was complete, several enzymes were not annotated, which did vary by clade. Amongst almost all assemblies, even the outgroup, the 3-keto steroid reductase, which catalyses the biosynthesis of zymosterol, was not observed in the KEGG annotation, which correlated to Belcour et al’s findings where zymosterol was not observed in C. crispus. Likewise, the cholestenol delta-isomerase orthologue K01824 was not observed in Florideophyte, Bangiophyte, or Porphyridiophyte algae, which supports Belcour et al’s findings that no delta7/8 iso-merase orthologues were found in C. crispus. Interestingly, 7-dehydrocholesterol re-ductase was not observed in the Cyanidiophyte algae, while it was annotated in most other assemblies. Likewise, delta24-sterol reductase was not annotated in the Galdier-iales, while cholestenol delata-isomerase was, indicating that Cyanidiophytes may possess a sterol biosynthesis pathway that is distinct from other Rhodophytes.”

To summarise: we discuss four particular enzymes: a 3-keto steroid reductase, a delta isomerase, and a 7-dehyrocholesterol reductase, and a delta24-sterol reductase. This supports Belcour’s findings that particular enzymes and metabolites are not in red seaweeds; however, some of the genes that are missing in red seaweeds are present in Cyanidiophytes. Also, thank you for your point on MetaCyc, which we will take on board.

3.

Reviewer: Similarly, expansion of animal-like peroxidases has already been reported in the genome of Chondrus crispus (page 5 of Collén et al., 2013, https://doi.org/10.1073/pnas.1221259110), so this paper should also be cited and discussed.

Response: We have added mention of this paper and its discussion, and cited it. We have also added a section into the results section: “This reflects the expansion in animal-heme peroxidases reported by Collén et al (2013) [55], which supports their assertion that it is an adaptation in red algae to life in marine environments where halogenated compounds are common and utilised in cellular de-fence and secondary metabolism. These adaptations have also been observed in brown algae, which also utilise halogenated metabolites in a similar manner [56]. It also suggests that this large expansion of heme-peroxidases was most pronounced in the Florideophytes, and not observed in other Rhodophyte classes.

Minor points

4.

Reviewer: lines 49-50:  "however red algae have been shown to possess phytosterols such as sitosterols, campesterol, and fucosterol" although in line with the cited reference, this seems to be a bold oversimplification. According to other authors, like Darnet et al, 2021 (doi:10.3389/fpls.2021.665206), there is variation in the stereochemistry of methyl groups on the lateral side chain, so it would be more prudent to tell that "red algae have been shown to posses small amounts of C24 methyl- and ethylsterols."

Response: We have incorporated the change as suggested.

5.

Reviewer: Lines 197: I am not sure it is meaningfull to separate phytosterol biosynthetic enzymes from cholesterol/ergosterol ones. Probably there is a strong overlap. For example the identical conservation pattern of K011853 an K011852 goes with the fact that lanosterol synthase and cycloartenol synthase are actually corresponding to the same orthogroup, lanosterol synthase being present only in animals and fungi. So Figure 2 could be simplified by grouping enzymes with similar catalytic activities and probably corresponding to same sequences.

Response: We have altered Figure 2 to combine all three pathways together. To improve readability and use, we then grouped each enzyme by type (e.g., synthases, monooxygenases) and colour coded. This also allows the reader to compare the annotation rates of different orthologues of the same type.

6.

Reviewer: Lines 256-257: "This contradicts in vitro evidence of sterol production in red algae, indicating that red algae may have novel genes involved in the biosynthesis of sterols that have not been identified in the KEGG database." This has been indeed demonstrated in diatoms (Pollier et al., 2018; https://doi.org/10.1038/s41564-018-0305-5).

Response: We have added this section in the results (2.2.1):

Evidence of alternative sterol biosynthesis genes have been found in other organisms. For example, alternative squalene epoxidase genes, which is key to steroid production in eukaryotes, have been shown to have alternative genes not previously identified in di-atoms and Cryptophyceae [38]. Furthermore, Belcour et al 2020 [39] describes an alternate cholesterol biosynthesis pathway for Chondrus crispus predicted using in silico analysis that differs from the pathway used by the KEGG database.”

7.

Reviewer: Lines 420-421: "A carrageenan biosynthesis pathway has been proposed [43]"

 a more up do date reference from the same group is Lipinska et al., 2020 (https://doi.org/10.1038/s41598-020-67728-6), where differencial expression of candidate enzymes is discussed.

Response: Section has been updated as follows (additions highlighted in red):

“A carrageenan biosynthesis pathway has been proposed by Ficko-Blean et al (2015) [44], and further explored by Lipinska et al (2020) [45], but does not yet have any orthologues on the KEGG database.”

8.

Reviewer: Various typos (not listed here for brevity).

Response: Typos have been fixed. Thank you for bringing these to our attention.

Reviewer 3 Report

Comments and Suggestions for Authors

This manuscript provided a Rhodophyte (red algae) multi-omics resource based on the publicly available data. Overall, it provides new and useful information on the metabolic profiles of available Rhodophyte assemblies and is presented in a very structure manner. Thus, it could be considered for publication in Marine Drugs. There are some suggestions in follows.

 1.     The sterol biosynthesis pathways (including possible alternative pathway) should be mapped for better understand how to improve the sterol production in Rhodophyte. The strategies to increase sterol production should be discussed.

2.     The innovation of this paper (compared to the related research) and its guiding significance for future work need to be clarified clearly in Introduction and Discussion.

3.  Figure 4C is not provided.

Author Response

1.

Reviewer: The sterol biosynthesis pathways (including possible alternative pathway) should be mapped for better understand how to improve the sterol production in Rhodophyte. The strategies to increase sterol production should be discussed.

Response: We believe that this request is outside the scope of this study. However, we have expanded discussions to refer to literature where a novel sterol biosynthesis study has been described by Belcour et al 2020, and discussed how our results align with that pathway. In the results we added this to section 2.2.1:

Evidence of alternative sterol biosynthesis genes have been found in other organisms. For example, alternative squalene epoxidase genes, which is key to steroid production in eukaryotes, have been shown to have alternative genes not previously identified in di-atoms and Cryptophyceae [38]. Furthermore, Belcour et al 2020 [39] describes an alternate cholesterol biosynthesis pathway for Chondrus crispus predicted using in silico analysis that differs from the pathway used by the KEGG database.”

And in the discussion section 3.4:

This evidence supports the findings of Belcour et al (2020) [39], which used a new bioinformatic tool to infer a new putative cholesterol biosynthesis pathway in C. crispus. This study found that while most of the cholesterol pathway was complete, several enzymes were not annotated, which did vary by clade. Amongst almost all assemblies, even the outgroup, the 3-keto steroid reductase, which catalyses the biosynthesis of zymosterol, was not observed in the KEGG annotation, which correlated to Belcour et al’s findings where zymosterol was not observed in C. crispus. Likewise, the cholestenol delta-isomerase orthologue K01824 was not observed in Florideophyte, Bangiophyte, or Porphyridiophyte algae, which supports Belcour et al’s findings that no delta7/8 isomerase orthologues were found in C. crispus. Interestingly, 7-dehydrocholesterol reductase was not observed in the Cyanidiophyte algae, while it was annotated in most other assemblies. Likewise, delta24-sterol reductase was not annotated in the Galdieriales, while cholestenol delta-isomerase was, indicating that Cyanidiophytes may possess a sterol biosynthesis pathway that is distinct from other Rhodophytes.

2.

Reviewer: The innovation of this paper (compared to the related research) and its guiding significance for future work need to be clarified clearly in Introduction and Discussion.

Response: The innovation in this paper and its significance is largely related to the scale of the work, and the inclusion of new omics data that has been published relatively recently. Thus, we have added the following section the introduction:

Furthermore, several important Rhodophytes that have historically not had much genomic investigation have had assemblies published recently. For example, the red seaweed Asparagopsis taxiformis, which is being researched as a potential anti-methanogenic compound only had a full genome assembly published in 2020 [14], while several assemblies from edible seaweeds including Gracilaria and Neoporphyra sp. were only published within the last five years.”

Furthermore, we have added a new discussion section (3.8) discussing significance of this study and future directions:

3.8 Future Rhodophyte innovation needs greater publication of high quality, annotated assemblies and greater metabolomic integration

This study used a broad bioinformatic approach to compare metabolic trends across Rhodophyte assemblies in both genomes and transcriptomes. However, the lack of annotated genome assemblies, and the relative low quality of several of these assemblies, hampered effective analysis. It has been reported that in some metrics, algal assembly quality has gone down, such as through the increased publication of fragmented genome assemblies without scaffolding [16]. Rhodophyte assemblies in particular also do not have a dedicated database for assembly quality testing with BUSCO, and typically show lower scores on average compared to green algae [16,63].

This study has also shown incomplete biosynthetic pathway annotations of key metabolites, including sterols and nucleotide biosynthesis pathways using the KEGG database, despite in vitro evidence of their production in red algae, indicating that novel genes or pathways are involved in these biosynthesis genes. This also contrasts to other data sources, such as the recent description of an algal cholesterol biosynthesis pathway from C. crispus [47]. Thus, this study suggests further research to develop our under-standing of Rhodophytes, and the development of bioinformatic tools and datasets de-signed for use with Rhodophyte assemblies, such as a BUSCO database, and for further modelling and biochemical analysis of algal metabolic pathways, including those which are otherwise well known in other organisms, and their inclusion into public databases used for functional annotation of assemblies.”

3.

Reviewer: Figure 4C is not provided.

Response: Changed labelling of Figure 4. Figure 4C now refers to the second tree, with 4B now referring to the bar chart of gene duplications.

Reviewer 4 Report

Comments and Suggestions for Authors

Rhodophytes have extensive research and application value, but genomic studies on Rhodophytes are relatively lagging behind. Reports on whole-genome and transcriptome data are limited, and most lack gene functional annotations. In this study, bioinformatics methods were applied to systematically analyze the genomic and transcriptomic data of Rhodophytes. These include the incomplete annotation pathway of the sterol biosynthesis pathway in all Rhodophytes and the highly duplicated phenomenon of halogen-related heme peroxidase genes in Florideophytes. These results are of significant importance for studying the biosynthesis of secondary metabolites in Rhodophytes. However, some questions require the author to provide reasonable explanations:

1. The study of gene functional annotation found that the sterol biosynthetic pathway in red algae is incomplete. However, previous studies have indicated the presence of sterol in Rhodophytes. To validate the data in this study, the author should test whether Rhodophytes contain sterol.

2. This paper discusses the genes involved in sterol synthesis pathways and duplication of halogen-related heme peroxidase genes. I think these discussions are too general. Could the author provide 1-2 specific cases to illustrate the practical application of bioinformatics in the analysis of biosynthetic pathways? For example, if the study shows that Rhodophytes lack genes in the sterol biosynthetic pathway, how can this phenomenon be explained based on this study?

3. Organisms often use gene duplication to achieve subfunctionalization of genes. The analysis results show the duplication of halogen-related heme peroxidase genes in Rhodophytes. How much useful information can readers obtain from this when studying the biosynthesis of secondary metabolites in Rhodophytes? Or how can readers extract effective information from it?

Author Response

1.

Reviewer: The study of gene functional annotation found that the sterol biosynthetic pathway in red algae is incomplete. However, previous studies have indicated the presence of sterol in Rhodophytes. To validate the data in this study, the author should test whether Rhodophytes contain sterol.

Response: We believe that there may have been a miscommunication of the results, which we have rectified through several adjustments, including the title, statements in the text, and several headers. The main result is not that the pathways themselves are incomplete, but rather that the annotations using the existing pathways are incomplete, thus suggesting that the sterol pathways are different to currently known ones. We have also added mention to a novel sterol pathway described by Belcour et al 2020. We have also added references to Belcour et al’s findings of whether specific compounds were detected, further adding weight to the prior references that various sterols are indeed biosynthesised by Rhodophytes.

2.

Reviewer: This paper discusses the genes involved in sterol synthesis pathways and duplication of halogen-related heme peroxidase genes. I think these discussions are too general. Could the author provide 1-2 specific cases to illustrate the practical application of bioinformatics in the analysis of biosynthetic pathways? For example, if the study shows that Rhodophytes lack genes in the sterol biosynthetic pathway, how can this phenomenon be explained based on this study?

Response: We believe this point has been covered by the alterations made in response to other points. To clarify, we have discussed in greater depth the sterol biosynthesis pathways, and compared the results regarding specific enzymes to the findings of Belcour et al 2020 (Discussion 3.4), and made it cleared that Rhodophytes are not missing genes in the sterol pathways, but rather that the previously published gene pathways for sterol biosynthesis do not adequately cover Rhodophytes. We have also brought more attention to gene subfunctionalisation in section 3.7, and addressed specific issues with current Rhodophyte omics in section 3.8.

In section 3.7:

This reflects the expansion in animal-heme peroxidases reported by Collén et al (2013) [55], which supports their assertion that it is an adaptation in red algae to life in marine environments where halogenated compounds are common and utilised in cellular de-fence and secondary metabolism. These adaptations have also been observed in brown algae, which also utilise halogenated metabolites in a similar manner [56]. It also suggests that this large expansion of heme-peroxidases was most pronounced in the Florideophytes, and not observed in other Rhodophyte classes.

3.

Reviewer: Organisms often use gene duplication to achieve subfunctionalization of genes. The analysis results show the duplication of halogen-related heme peroxidase genes in Rhodophytes. How much useful information can readers obtain from this when studying the biosynthesis of secondary metabolites in Rhodophytes? Or how can readers extract effective information from it?

Response: We have added some discussion to gene subfunctionalisation. To set up this discussion, we added a section into the introduction:

The biosynthesis of halogenated metabolites is reliant on numerous enzymatic reactions and biosynthetic genes. Of these, heme peroxidases and related enzymes are well known to be important enzymes in the biosynthesis of halogenated metabolites [15], and have been observed to be duplicated in C. cripsus [16]. Gene duplication is a major contributor to evolution, and is a known source of gene subfunctionalisation [17].

We then added more into the discussion section 3.7 (additions highlighted in red):

Haem peroxidases form a wide range of enzymes that are involved in oxidation reactions utilising hydrogen peroxide as an oxygen acceptor, and are involved in the biosynthesis of various metabolites, including secondary metabolites in plants and algae, as well as other cellular functions such as cell signalling and defence [57-59]. Further analysis of the annotations of these enzymes indicates that this gene duplication may have resulted in the subfunctionalisation of these genes. The majority of heme peroxidases inferred to be duplicated were annotated as animal heme peroxidases, which is an old denomination yet still pre-sent in some annotation databases, and is associated with the oxidation of halides [59], while others were annotated as Peroxinectin; a multifunctional molecule with adhesive and peroxidase activity [60]. Both belong to the peroxidase-cyclooxygenase superfamily [59,60]. G. lemaneiformis and G. chorda also had several genes annotated as eosinophil peroxidase, a heme peroxidase involved in the oxidation of bromide ions [61]. However, further research would have to be performed to fully explore how gene du-plication was involved in the subfunctionalisation of heme peroxidases in Rhodophytes. Given that Florideophyte algae are known for their production of brominated natural products that rely on the production of hydrogen peroxide in their biosynthesis [14,62], the duplication of halide-associated haem peroxidases highlights the evolution of to-wards the production of halogenated secondary metabolites in red seaweeds, and in particular towards the utilisation of brominated compounds in Florideophyte seaweeds.”

In short, finding further details of subfunctionalisation requires additional investigation of other annotations, which we briefly discuss in the text. However, in-depth analysis of subfunctionalisation would require further research to adequately explain. Furthermore, due to the interrelated nature of secondary metabolic pathways in plants and their general lack of specific pathway annotations, we have not been able to link duplication of heme peroxidases with specific metabolites.

Round 2

Reviewer 2 Report

Comments and Suggestions for Authors

The paper has been significantly improved, but I still have some minor comments.

Regarding genomic cross-checking with previous studies, authors should also cite Desmond and Gribaldo, 2009 (DOI: 10.1093/gbe/evp036) who performed the first analysis of candidate enzymes for sterol metabolism in O. tauri and C. merolae. 

Figure 2, page 6: 

I wonder if it is meaningful to maintain K01852 on the figure given that lanosterol has not been reported in red algae. All sequences identified in the study could probably be annotated as cycloartenol synthases (K01853). I also do not understand how it is possible that the sequence from C. kondoi is annotated only as lanosterol synthase whereas the one from G. turuturu is annotated only as cycloartenol synthase? I would expect automated annotation not being able to differenciate between both. 

Regarding K14423, K14424, K07750, all three could be referred as C-4alpha-methyl-monooxygenase. Given that K07750 is the more general version of K14423 and K14424, it is also surprizing that some sequences assigned to K14423 and/or K14424 are not also assigned to K07750. This would also need a little bit more exploration. 

Also K07748 sterol-4alpha-carboxylate 3-dehydrogenase (decarboxylase) and K09827 3-keto steroid reductase should appear just after in the table, it makes more biochemical sense because those two reaction are part of the tree-step C4-demethylation (see Desmond2009 for example).

K05917 could be more precisely referred in the legend as 14alpha-demethylase. I wonder if the authors tried to perform a tblastn on C. paradoxa genome to check if the gene is really absent or just not predicted as a protein-coding sequence?

As discussed in the text, the distribution of sterol 24-C-methyltransferase is globally consistent with chemical profiling data (Kodner et al. 2008 10.1111/j.1472-4669.2008.00167.x) indicating that unicellular red alga do have methylsterols. But this also makes somewhat surprizing the reports of methyltransferases in N. yezoensis ZULJ, G. filicina and E. denticulatum. Could it be that those are contaminations? Blasting them against other eukaryotic reference sequences could give a first idea about this, and the case could be resolved by comparative genomics if there is remaining ambiguity. 

The reductase part should be reorganized according to related catalytic activities:

K00223 and K09828 (delta24-reductases) should be discussed together. Once again it's disturbing to notice that the analysis pipeline gets so different results for those two. This should be discussed.  

The same remark applies (to a lesser extent) to K00222 and K19532 (Delta14-sterol reductases).

Regarding K08246 (cycloisomerase), the rare losses are probably due to annotation issues, this should be cross-checked with tblastn. Morover, it is unrelated to the two delta8-isomerases (K01824 and K09829), so should be analyzed separately.

Regarding K09829 (C-8 sterol isomerase), kegg taxonomy indicates this gene is present in C. crispus, as reported also in Belcour2020. It is therefore surprizing that it is absent in the analysis, and should be discussed. 

The two distinct desaturases, K00227 (Delta7-sterol 5-desaturase) and K09831 (C22-desaturase) should also be analyzed separately. 
